# Distinct Mechanistic Behaviour of Tomato CYP74C3 and Maize CYP74A19 Allene Oxide Synthases: Insights from Trapping Experiments and Allene Oxide Isolation

**DOI:** 10.3390/ijms24032230

**Published:** 2023-01-23

**Authors:** Alexander N. Grechkin, Natalia V. Lantsova, Lucia S. Mukhtarova, Bulat I. Khairutdinov, Svetlana S. Gorina, Tatiana M. Iljina, Yana Y. Toporkova

**Affiliations:** Kazan Institute of Biochemistry and Biophysics, FRC Kazan Scientific Center of RAS, P.O. Box 261, 420111 Kazan, Russia

**Keywords:** cytochrome P450, CYP74A, CYP74C, allene oxide synthase mechanism, product specificity, allene oxide structure

## Abstract

The product specificity and mechanistic peculiarities of two allene oxide synthases, tomato LeAOS3 (CYP74C3) and maize ZmAOS (CYP74A19), were studied. Enzymes were vortexed with linoleic acid 9-hydroperoxide in a hexane–water biphasic system (20–60 s, 0 °C). Synthesized allene oxide (9,10-epoxy-10,12-octadecadienoic acid; 9,10-EOD) was trapped with ethanol. Incubations with ZmAOS produced predominantly 9,10-EOD, which was converted into an ethanolysis product, (12*Z*)-9-ethoxy-10-oxo-12-octadecenoic acid. LeAOS3 produced the same trapping product and 9(*R*)–α–ketol at nearly equimolar yields. Thus, both α–ketol and 9,10-EOD appeared to be kinetically controlled LeAOS3 products. NMR data for 9,10-EOD (Me) preparations revealed that ZmAOS specifically synthesized 10(*E*)-9,10-EOD, whereas LeAOS3 produced a roughly 4:1 mixture of 10(*E*) and 10(*Z*) isomers. The cyclopentenone *cis*-10-oxo-11-phytoenoic acid (10-oxo-PEA) and the Favorskii-type product yields were appreciable with LeAOS3, but dramatically lower with ZmAOS. The 9,10-EOD (free acid) kept in hexane transformed into macrolactones but did not cyclize. LeAOS3 catalysis is supposed to produce a higher proportion of oxyallyl diradical (a valence tautomer of allene oxide), which is a direct precursor of both cyclopentenone and cyclopropanone. This may explain the substantial yields of *cis*-10-oxo-PEA and the Favorskii-type product (via cyclopropanone) with LeAOS3. Furthermore, 10(*Z*)-9,10-EOD may be produced via the reverse formation of allene oxide from oxyallyl diradical.

## 1. Introduction

Allene oxide synthase (AOS, syn. hydroperoxide dehydratase, EC 4.2.1.92) is an enzyme controlling the dehydration of fatty acid hydroperoxides to the short-lived allene oxides [1,2]. AOSs of the CYP74 family (P450 superfamily) are widespread in plants and play a key role in biosynthesis of phytohormone jasmonates [3,4]. AOSs of stony corals are also P450s of the CYP74 clan but not the CYP74 family [5]. AOSs detected in some fungi are distinct P450s, having no relation to the CYP74 family and clan [6]. Another kind of AOS present in some soft corals is the catalase-like haemoproteins of the peroxidase superfamily, not the P450s [2].

Although the primary products of AOSs are short-lived allene oxides, the total conversion is rather complex. Firstly, the allene oxides themselves are known to co-exist with their valence tautomers oxyallyl and cyclopropanone [7]. Secondly, the *exo*-double bond at the oxirane of allene oxide may have either (*E*) or (*Z*) configuration depending on the AOS specificity [8,9,10]. Thirdly, the short-lived allene oxides undergo rapid conversions into ketols (hydrolysis) [11,12] or cyclopentenones (cyclization) [13,14,15]. Finally, some AOSs, such as those of the CYP74C subfamily, possess distinct product patterns compared to the most common CYP74A subfamily AOSs. For instance, the CYP74C AOSs produce substantial yields of cyclopentenone *cis*-10-oxo-11-phytoenoic acid (*cis*-10-oxo-PEA) via the corresponding allene oxide [10,16,17,18]. In contrast, the CYP74A enzymes produce essentially no cyclopentenones from linoleic acid hydroperoxides via allene oxides [19,20,21]. This difference was previously attributed to either a geometric specificity of CYP74C AOSs, producing the “cyclizable” allene oxide isomer [10], or the AOS-assisted allene oxide cyclization [18]. Overall, although allene oxide fatty acids were discovered in 1987, there are still some unsolved questions regarding the mechanistic peculiarities of different AOSs, including the specificity of allene oxide synthesis and conversions. This prompted us to conduct the present comparative study of mechanistic behaviour of two distinct AOSs, namely the tomato CYP74C3 (LeAOS3) and maize CYP74A19 (ZmAOS). The results of experiments, including the brief biphasic incubations, trapping experiments, isolation and identification of products, including the allene oxides, are described in the present paper. The ethanol trapping experiments and the NMR structural studies of allene oxides provided new insights into the specificities and mechanistic peculiarities of short-lived allene oxides and cyclopropanone formation and their subsequent conversions.

## 2. Results

### 2.1. Ethanol Trapping Experiments with 9-HPOD. Major Products

To minimize the hydrolysis of allene oxide, the incubations (described in full detail in Materials and Methods) were carried out in a biphasic system via a modified procedure by Brash et al. [10]. An ice-cold solution of linoleic acid 9(*S*)-hydroperoxide (9-HPOD) in hexane was extensively vortexed with LeAOS3 or ZmAOS for 20 s–5 min (as specified below). The water was quickly frozen at −79 °C, the hexane phase was decanted, and an excess of ice-cold ethanol was directly added to the hexane solution for allene oxide trapping. The predominant part of hexane was evaporated in vacuo and carboxylic acids were esterified with ethereal diazomethane. The resulting Me esters were trimethylsilylated and, thus, free alcohol functions present in some products were converted into TMS derivatives. The products were subjected to GC-MS analyses. The structural formulae of the major products described below are presented in Figure 1. 

The profiles of ethanol trapping products formed after 9-HPOD incubation with LeAOS3 and ZmAOS for 60 s are shown in Figure 2A,B, respectively. These two profiles were significantly distinct. Trapping after ZmAOS incubation (Figure 2B) resulted in a single predominant peak **1** (Figure 1). Compound **1** is the product of the ethanolysis of allene oxide (9,10-epoxy-10,12-octadecadienoic acid; 9,10-EOD), as judged by the spectral data described below. In contrast, the LeAOS3 trapping profile (Figure 2A) possessed two nearly equal major peaks of the same product **1** and α–ketol (**2**, Me/TMS). Furthermore, the LeAOS3 profile (Figure 2A) possessed a perceptible peak of cyclopentenone *cis*-10-oxo-11-phytoenoic acid (**4**, 10-oxo-PEA, Me), which was absent from the ZmAOS profile (Figure 2B). The mass spectra for α–ketol (**2**, Me/TMS) and 10-oxo-PEA (**4**, Me) are not illustrated since they fully corresponded to those described before [18].

The mass spectrum of the predominant product **1** (Me) is presented in Figure 2C. The spectrum possessed M^+^ at *m*/*z* 354 (0.1%) and [M – MeO]^+^ at *m*/*z* 323 (1%). Fragment [M – Me(CH_2_)_7_CH=O]^+^ at *m*/*z* 215 (100%) indicated the presence of 9-oxo-10-ethoxy function. The mass spectrum of the product of catalytic hydrogenation of compound **1** (Me) and the corresponding mass fragmentation scheme are presented in Appendix A. The spectrum possessed M^+^ at *m*/*z* 356 (0.1%) and [M – MeO]^+^ at *m*/*z* 325 (2%), thus, indicating the addition of two hydrogens upon hydrogenation. The rest of the spectrum was nearly identical to that of compound **1**. The NaBH_4_ reduction of compound **1** followed by the sequential methylation/trimetylsilylation resulted in a product whose mass spectrum (Appendix A) possessed M^+^ at *m*/*z* 428 (0.1%), [M – Me]^+^ at *m*/*z* 413 (0.3%) and [M – MeO]^+^ at *m*/*z* 397 (2%). Complementary fragments at *m*/*z* 213 and *m*/*z* 215 (see the mass fragmentation scheme, Appendix A, inset) signified the presence of *vic*-diol function at C9/C10. The fragment at *m*/*z* 317 and the losses thereof indicated the TMSO and EtO substituents at C10 and C9, respectively. Overall, the spectrum indicated the structure of (12*Z*)-9-ethoxy-10-hydroxy-12-octadecenoic acid (Me/TMS) for NaBH_4_ reduced compound **1**. In turn, the obtained data allow one to ascribe the structure of (12*Z*)-9-ethoxy-10-oxo-12-octadecenoic acid (Me), i.e., the ethanolysis product of 9,10-EOD, to parent compound **1**.

The α–ketol **2** formed upon the brief (60 s) incubations of LeAOS3 with 9-HPOD followed by ethanol trapping was isolated and purified by NP-HPLC. Purified α–ketol was subjected to the analysis of its enantiomeric composition by chiral-phase HPLC. The obtained results showed that α–ketol was composed of 92% pure (9*R*) enantiomer. The isomer was not detectable. In contrast, the same α–ketol **2** formed after the identical incubation with ZmAOS followed by ethanol trapping was composed of ca. 70% (9*R*) and 30% (9*S*) enantiomers.

A minor product (retention time 16.08 min) detected after the ethanol trapping experiments with LeAOS3 was eluted in front of the *cis*-10-oxo-PEA (**4**) peak (Figure 2A). The mass spectral patterns of product **3** (Et/Me), shown in Appendix A, closely matched those of the dimethyl ester of the Favorskii-type products previously described [18]. The spectrum possessed [M – MeO]^+^ at *m*/*z* 323 (2%), [M – EtOH]^+^ at *m*/*z* 308 (4%) and other distinctive fragments (see the Appendix A). The mass fragmentation indicated the structure of (2′*Z*)-2-(2′-octenyl)-decane-1,10-dioic acid (Et/Me ester) formed via the ethanolysis of short-lived cyclopropanone. For further structural confirmation, product **3** was hydrogenated over PtO_2_. The mass spectrum of hydrogenated product (Appendix A) showed characteristic fragmentation patterns (inset in the same figure). Most of the fragments were formed due to the cleavage of bonds at tertiary carbon (C2). The results enabled us to assign the structure of 2-(octyl)-decane-1,10-dioic acid (Et/Me ester) to hydrogenated compound **2**, thereby confirming the parent compound **3** structure as (2′*Z*)-2-(2′-octenyl)-decane-1,10-dioic acid (Et/Me ester). Compound **3** was also detected after the ethanol trapping experiments with ZmAOS, but it was ca. 100-times less abundant compared to the LeAOS3 ethanol trapping products.

### 2.2. Ethanol Trapping Experiments with 13-HPOT. Major Products

A biphasic incubation of ZmAOS with α−linolenic acid 13(*S*)-hydroperoxide (13-HPOT) followed by ethanol trapping resulted in the formation of a single predominant product **5** (Figure 3B). The mass spectrum of product **5** (Me) and its mass fragmentation scheme are presented in Figure 3C and its inset, respectively. Compound **5** (Me) exhibited M^+^ at *m*/*z* 352 and a base peak [M – C1/C12]^+^ at *m*/*z* 127, was identified as the expected allene oxide ethanolysis product (Me); see the spectrum and the mass fragmentation scheme in Figure 3C and its inset, respectively. The mass spectrum (Appendix A) of the product of compound **5** hydrogenation possessed [M – C1/C12]^+^ at *m*/*z* 129 (100%) and [129 – EtOH]^+^ at *m*/*z* 83 (58.9%). These spectral peculiarities, as shown in the fragmentation scheme (Appendix A, inset), advocated the structure of 12-oxo-13-ethoxyoctadecanoic acid (Me) for hydrogenated product **5**. The product of NaBH_4_ reduction of compound **5**, followed by methylation and trimethylsilylation, exhibited the mass spectral profile shown in Appendix A. The spectrum possessed M^+^ at *m*/*z* 426 (0.04%), [M – Me]^+^ at *m*/*z* 411 (0.51%), [M – MeO]^+^ at *m*/*z* 395 (2%) and other characteristic fragments, as shown in the fragmentation scheme (Appendix A, inset). Overall, the obtained data confirmed the structure of (9*Z*,15*Z*)-13-ethoxy-12-oxo-9,15-octadecadienoic acid (Me), i.e., the ethanolysis product of allene oxide (12,13-epoxy-9,11,15-octadecatrienoic acid; 12,13-EOT) for compound **5**. The α–ketol, (9*Z*,15*Z*)-12-oxo-13-hydroxy-9,15-octadecadienoic acid (Me/TMS), was present, but it was in a minority compared to product **5**. Another minor product was identified as *cis*-12-oxo-10,15-phytodienoic acid (12-oxo-PDA, **8**) Me ester (Figure 3B).

The less-polar minor product **7** possessed mass spectral patterns similar to those of the previously described [21] Favorskii-type product (dimethyl ester). The spectrum possessed [M – MeO]^+^ at *m*/*z* 321 (1%), [M – EtOH]^+^ at *m*/*z* 306 (7%), [M – EtCH=CHCH_2_]^+^ at *m*/*z* 283 (6%), [M – HCOOEt]^+^ at *m*/*z* 278 (21%) and [321 – EtOH]^+^ at *m*/*z* 275 (19%). The full spectrum for product **7** and the corresponding mass fragmentation scheme (inset) are presented in Appendix A. The spectral data advocated the structure of (2′*Z*,4*Z*)-2-(2′-pentenyl)-4-tridecene-1,13-dioic acid (Et/Me) for compound **7**. Catalytic hydrogenation of product **7** over PtO_2_ produced a saturated analogue possessing a distinctive base peak [M – (CH_2_)_10_COOMe + H]^+^ at *m*/*z* 158 (100%) due to the fragmentation at the branching point. The full spectrum for hydrogenated product **7** and the corresponding mass fragmentation scheme (inset) are presented in the Appendix A. The spectral data were consistent with the structure of (2′-pentyl)-4-tridecane-1,13-dioic acid (Et/Me) for hydrogenated product and (2′*Z*,4*Z*)-2-(2′-pentenyl)-4-tridecene-1,13-dioic acid (Et/Me) for the parent compound **7**. This structure corresponds to the Favorskii-type rearrangement that occurred through the ethanolysis of cyclopropanone, a valence tautomer of allene oxide, 12,13-EOT.

Similar LeAOS3 incubations with 13-HPOT followed by ethanol trapping also resulted in the formation of compound **5** (Me), but at a lower yield than after the above-described ZmAOS incubations (Figure 3A compared to Figure 3B). Unlike in the case of ZmAOS, LeAOS3 trapping experiments resulted in a higher yield of α–ketol (**6**, Me/TMS), which possessed the following mass spectral patterns: M^+^ at *m*/*z* 396 (0.02%), [M – Me]^+^ at *m*/*z* 381 (2%), [M – MeO]^+^ at *m*/*z* 365 (1%), [M – EtCH=CHCH_2_]^+^ at *m*/*z* 327 (0.4%), [M – C12/C18 + TMS]^+^ at *m*/*z* 270 (18%), [M – C1/C12]^+^ at *m*/*z* 171 (100%), *m*/*z* 129 (45%), [CH_2_=O^+^–SiMe_3_] at *m*/*z* 103 (22%), *m*/*z* 81 (44%) and [Me_3_Si]^+^ at *m*/*z* 73 (81%). The cyclopentenone *cis*-12-oxo-PDA (Me) was detected as a minority (Figure 3A).

Incubation of both LeAOS3 and ZmAOS with linoleic acid 13(*S*)-hydroperoxide (13-HPOD) resulted in poorer yields of products. Nevertheless, an allene oxide trapping product **9**, (9*Z*)-12-oxo-13-ethoxy-9-octadecenoic acid (Me) was detected. These results are described in the Supplementary Information.

### 2.3. Allene Oxide Isolation and NMR Study

The allene oxide 9,10-EOD (Me) was prepared after biphasic incubations of ZmAOS and LeAOS3 with 9-HPOD followed by methylation with diazomethane and purification by NP-HPLC on a LiChrosphere CN (5 μm) 30 × 4 mm column maintained at –15 °C. The NMR spectra (^2^H_14_-hexane, 253 K) were recorded. The yield achieved after the ZmAOS incubation allowed us to record the ^1^H-NMR, ^1^H-^1^H-COSY, ^1^H-^1^H-TOCSY, ^1^H-^1^H-NOESY, ^1^H-^13^C-HSQC and ^1^H-^13^C-HMBC spectral data, presented in Table 1. The ^1^H-^13^C-HSQC and ^1^H-^13^C-HMBC are also depicted in Appendix A, and the ^1^H-^1^H-COSY spectrum is shown in Appendix A. All signal attributions were verified by COSY, TOCSY and HMBC correlations. The spectrum matched that previously described for allene oxides produced by CYP74A AOSs [9,10]. The chemical shifts in all three olefinic protons, H11, H12 and H13 (Table 1, Figure 4), were in agreement with the (*E*)-configuration of the *exo* double bond at the oxirane (C10) [10]. Overall, the NMR data (Table 1) confirmed the structure of (10*E*,12*Z*)-9,10-epoxy-10,12-octadecadienoic acid (Me) for 9,10-EOD biosynthesized by ZmAOS.

LeAOS3 produced 9,10-EOD at a much poorer yield. Nevertheless, the 9,10-EOD (Me) was also isolated, purified by HPLC and the ^1^H-NMR, ^1^H-^1^H-COSY and ^1^H-^1^H-TOCSY spectra were recorded (Table 2). The ^1^H-^1^H-NOESY, ^1^H-^13^C-HSQC and ^1^H-^13^C-HMBC spectra were not recorded due to the too-low 9,10-EOD concentration. The ^1^H-NMR spectrum (Figure 4B) revealed the presence of two geometric isomers of 9,10-EOD at a ratio of ca. 4:1, as judged by the signal integration data. The major isomer signals matched those of the above-described 10(*E*)-isomer synthesised by ZmAOS. Two isomers were distinguished by their ^1^H-^1^H-COSY and ^1^H-^1^H-TOCSY correlations. The minor isomer had very similar shapes of olefinic proton multiplets to those of the 10(*E*)-isomer, but their chemical shifts were significantly different (Figure 4). For instance, the H12 signal was downshifted from 5.87 ppm (*E*-isomer) to 6.11 ppm (*Z*-isomer) due to the deshielding effect of epoxide oxygen. Contrariwise, the H11 signal was upfield shifted from 5.66 ppm (*E*-isomer) to 5.49 ppm, since H11 of the minor *Z*-isomer is anti-configured towards the oxygen atom of oxirane and is, thus, less affected by oxygen anisotropy. The chemical shifts in the minor isomer matched those of the (11*Z*)-isomer described before [10]. Overall, the recorded NMR data demonstrated that ZmAOS specifically synthesized (10*E*)-9,10-EOD, while LeAOS3 produced two geometric isomers of allene oxide, namely (10*E*)-9,10-EOD and (10*Z*)-9,10-EOD, at a ratio ca. 4:1.

To shed more light on the mechanistic specificities of two enzymes, the kinetics and concentration dependencies were studied. The results are described in the next section.

### 2.4. The Relative Yields of Separate Products

Table 3 summarises rough quantitative data on the relative yields of fatty acid hydroperoxide conversion products by LeAOS3 and ZmAOS. The most obvious difference is the much higher yield of α–ketols upon LeAOS3 incubations with all substrates. The Favorskii-type product **3** from 9-HPOD was formed with LeAOS3 only, not ZmAOS. The ratio of the Favorskii-type product **3** to the allene oxide trapping product **1** after LeAOS3 incubations was nearly constant, irrespective of enzyme concentration and incubation time (results not presented). The conversion of 13-HPOT by both enzymes yielded the allene oxide trapping product **5** as well as the detectable Favorskii-type product **7**. The cyclopentenone *cis*-12-oxo-PDA (**8**) was formed, but at a lower yield compared to that of 10-oxo-PEA formed upon the LeAOS3 incubations with 9-HPOD.

The relative yield of cyclopentenone *cis*-10-oxo-PEA (**4**) in relation to α–ketol (**2**) was nearly constant, about 1:20 (as estimated by integration of TIC and selected ion chromatograms), irrespective of LeAOS3 concentration (25 µg, 50 µg, 100 µg, 150 µg or 200 µg). Furthermore, when the allene oxide (free acid) was kept for hours in hexane solution, the ratio of *cis*-10-oxo-PEA (**4**) in relation to α–ketol (**2**) remained about the same. Incubations of LeAOS3 with 9-HPOD for 20, 40 or 75 s in a biphasic system followed by EtOH trapping resulted in product profiles possessing nearly the same constant proportion of *cis*-10-oxo-PEA (**4**) to α–ketol (**2**).

### 2.5. Allene Oxide Conversions in Aprotic Solvent

A special series of experiments was conducted to elucidate the allene oxide conversions in an aprotic solvent. Allene oxide, 9,10-EOD (free carboxylic acid), prepared by the biphasic incubations of LeAOS3 and ZmAOS with 9-HPOD, was allowed to stay in the hexane solution. Then, the products were treated with diazomethane, trimethylsilylated and subjected to GC-MS analyses. The resulting profiles of products are shown in Figure 5. Incubations with both LeAOS3 and ZmAOS resulted in the formation of two nonpolar products **11** and **12** (Figure 5A,B). The profile of LeAOS3 incubation products possessed α–ketol as a major constituent (Figure 5A). In contrast, analogous ZmAOS incubation yielded mainly the products **11** and **12** (Figure 5B), while α–ketol was only a minority. Compounds **11** and **12** were formed at a ratio of about 2:1 (by total ion current) with LeAOS3 and 2:3 with ZmAOS. The mass spectra and fragmentation schemes for products **11** and **12** presented in Figure 6 corresponded to those previously described for macrolactones (12*Z*)-10-oxo-12-octadecen-11-olide and (12*Z*)-10-oxo-12-octadecen-9-olide, respectively [16]. Catalytic hydrogenation over PtO_2_ converted products **11** and **12** into the saturated analogues, 10-oxooctadecan-11-olide and 10-oxooctadecan-9-olide (Appendix A). Compounds **11** and **12** are the products of intramolecular nucleophilic substitution, specifically, the carboxylic group attack on the oxirane of allene oxide.

When an allene oxide (12,13-EOT, free acid) prepared by biphasic incubation of 13-HPOT with LeAOS3 or ZmAOS was analogously kept in hexane solution, it yielded a single nonpolar product **13**, whose mass spectrum exhibited M^+^ at *m*/*z* 292 (0.8%), [M – Et]^+^ at *m*/*z* 263 (1%), [M – CH_2_CH=CHEt]^+^ at *m*/*z* 223 (4%), *m*/*z* 210 (2%), [M – CO(CH_2_)_2_CH=CHEt]^+^ at *m*/*z* 181 (6%), *m*/*z* 163 (4%), *m*/*z* 145 (5%), *m*/*z* 135 (26%), [M – C1/C11]^+^ at *m*/*z* 111 (20%), *m*/*z* 93 (16%), [M – C1/C12]^+^ at *m*/*z* 83 (100%), *m*/*z* 69 (29%). Hydrogenation of compound **13** over PtO_2_ resulted in a saturated analogue, whose mass spectrum possessed M^+^ at *m*/*z* 296 (2%), [M – (CH_2_)_4_Me + H]^+^ at *m*/*z* 226 (2%), [M – (CH_2_)_5_Me]^+^ at *m*/*z* 211 (2%), [M – CO(CH_2_)_5_Me]^+^ at *m*/*z* 183 (100%), [M – C1/C11]^+^ at *m*/*z* 113 (61%), *m*/*z* 95 (51%), [M – C1/C12]^+^ at *m*/*z* 85 (80%). The spectrum corresponded to that described before [22] and advocated the structure of 12-oxooctadecan-11-olide for the hydrogenation product. Overall, the obtained data uncovered the structure of (9,15)-12-oxo-9,15-octadecadien-11-olide for compound **13**.

An analogous macrolactone **14**, (9*Z*)-12-oxo-9-octadecen-11-olide, was detected when the allene oxide, 12,13-EOD, prepared by biphasic incubation of 13-HPOD with LeAOS3 or ZmAOS, was kept in hexane solution. The mass spectrum of compound **14** (not presented) exactly matched that described previously [22].

## 3. Discussion

### 3.1. General Aspects of Product Specificities of LeAOS3 and ZmAOS

Allene oxides (Me) produced by both ZmAOS (CYP74A19) and LeAOS3 (CYP74C3) were isolated and purified by NP-HPLC. ZmAOS provided a sufficient yield of allene oxide. The results of the NMR study demonstrated that ZmAOS (CYP74A19) produced the 10(*E*) isomer of allene oxide (9,10-EOD) quite specifically. The isolation of allene oxide produced by LeAOS3 was more complicated, largely due to the competitive formation of α–ketol (**2**). Thus, the yield of allene oxide was significantly lower compared to ZmAOS. Nevertheless, the allene oxide produced by LeAOS3 was isolated. The NMR study revealed the presence of two geometrical isomers of allene oxide, in full agreement with previous work [10]. The ratio of the 10(*E*) and 10(*Z*) isomers of 9,10-EOD detected was approximately 4:1. Thus, two enzymes possessed different geometrical specificities. A summarized scheme of allene oxide (9,10-EOD) synthesis and conversions by LeAOS3 and ZmAOS is presented in Figure 7.

In the present work, we used the approach of biphasic incubations of AOSs with substrates originally employed by Alan Brash [23], increasing the allene oxide yield due to the improved protection from hydrolysis. The combination of biphasic incubations with direct ethanol trapping of allene oxide without hexane evaporation enabled the efficient ethanolysis of short-lived allene oxide. Ethanol was used instead of methanol, which was originally used by Mats Hamberg [11], due to its free mixability with hexane. The obtained results showed that biphasic incubations followed by direct ethanol trapping are quite useful for allene oxide detection in reaction mixtures.

The ethanol trapping experiments with LeAOS3 and ZmAOS revealed distinct product specificities for both enzymes. Trapping with both LeAOS3 and ZmAOS expectedly yielded the products of allene oxide ethanolysis. On the other hand, two enzymes behaved differently in several respects. Firstly, LeAOS3, unlike ZmAOS, stereospecifically produced (9*R*)-α–ketol at a high yield during the brief incubations. Secondly, LeAOS3 produced a detectable yield of *cis*-10-oxo-PEA (**4**), which was essentially absent with ZmAOS. Thirdly, a perceptible peak of the Favorskii-type product **3** (Et/Me) was present after the LeAOS3 trapping experiments but nearly absent after analogous ZmAOS experiments. Detection of the Favorskii-type products indicated the presence of the short-lived cyclopropanones along with allene oxides. The Favorskii-type products (Et/Me) were formed via the ethanolysis of cyclopropanones (Figure 7), the valence tautomers of allene oxide.

### 3.2. Favorskii-Type Product Formation

The Favorskii-type product (Et/Me ester) (**3**) was consistently detected after LeAOS3 incubations with 9-HPOD followed by ethanol trapping. In contrast, the yield of product **2** was dramatically lower after analogous incubations of ZmAOS with 9-HPOD (Figure 7). The ratio of the Favorskii-type product (Et/Me ester) **3** to the 9,10-EOD trapping product, 9-ethoxy-10-oxo-12-octadecenoic acid (**1**), was nearly constant, irrespective of enzyme concentration and incubation time (data not presented).

The ethanol trapping performed after both LeAOS3 and ZmAOS incubations with 13-HPOT resulted in detectable yields of the Favorskii-type product **7** (Et/Me), in addition to the allene oxide trapping product **5** (Appendix A). Products **7** and **5** were formed through the ethanolysis of cyclopropanone and allene oxide, respectively. This indicated the co-existence of cyclopropanone with allene oxide (Appendix A). In addition to the Favorskii-type product **7**, *cis*-12-oxo-PDA (**8**) was detectable at a low yield after incubations with 13-HPOT. Thus, only a small part of the cyclizable allene oxide 12,13-EOT undergoes cyclization under these conditions. In contrast, in experiments with LeAOS3 (but not ZmAOS), the 9,10-EOD was quickly cyclized.

The expected allene oxide ethanolysis product **9** was also detected after similar experiments with 13-HPOD. At the same time, only traces of the Favorskii-type product, analogous to compound **7**, and no cyclopentenone were detectable after the trapping experiments after LeAOS3 or ZmAOS incubations with 13-HPOD (Appendix A). Thus, the formation of cyclopropanone showed a dependence on the β,γ double bond at the oxygen-bearing carbon of hydroperoxide, in agreement with the previously published data [21].

### 3.3. Cyclopentenone Formation

Allene oxides are able to cyclize spontaneously. However, the natural allene oxides, generated enzymatically, exist in the presence of water. Thus, their cyclization competes with hydrolysis. Most allene oxides, e.g., those synthesized from the linoleic acid hydroperoxides, are quickly hydrolysed to α–ketols and afford only traces of cyclopentenones. There are two exceptions. The first is exemplified by allene oxides, such as 12,13-EOT, which have an additional double bond in the β,γ position towards the oxirane [20,24]. Their cyclization capability was attributed to the anchimeric assistance of this double bond, which promotes oxirane opening and the formation of oxyallyl species, which are essential for cyclization [20,24]. The second exception is represented by CYP74C AOSs. The capability of biosynthesizing *cis*-10-oxo-PEA is a unique property of CYP74C AOSs such as CYP74C3 [10,16,17,18]. The most common AOSs of the CYP74A subfamily, such as ZmAOS, do not afford any noticeable yields of cyclopentenones from linoleic acid hydroperoxides [19,20]. Previously, the capability of LeAOS3 to produce 10-oxo-PEA was attributed to either (a) the formation of presumably cyclizable allene oxide (10*Z*)-9,10-EOD [10] or (b) control of allene oxide cyclization by LeAOS3 itself [18].

Observations from the present work allow one to presume that CYP74C AOSs such as LeAOS3 produce a tautomeric pair, allene oxide–oxyallyl, where the balance is partly shifted in favour of the latter. Both cyclopropanone and cyclopentenone ring closures are accomplished via the oxyallyl diradical (Figure 7). Both the Favorskii-type product (formed through the cyclopropanone) and 10-oxo-PEA production from 9-HPOD in the presence of LeAOS3 (but not ZmAOS) were detected in the present work. Their formation is a distinctive property of LeAOS3 compared to ZmAOS.

The cyclization of allene oxides to cyclopentenones was originally supposed to proceed via the conrotatory electrocyclization of a pericyclic pentadienyl cation [13,25,26]. However, recent theoretical studies have revealed that the cyclization occurs via the pericyclic pentadienyl diradical (oxyallyl diradical) [24,27,28,29,30,31]. The Favorskii-type product detected after incubating LeAOS3 with 9-HPOD is evidently formed via cyclopropanone (Figure 7). In turn, the closure of the cyclopropanone 3-membered ring proceeds through the oxyallyl diradical [24]. Therefore, these results indicate the occurrence of oxyallyl diradical in 9,10-EOD conversions in the presence of LeAOS3. The elevated cyclopentenone (*cis*-10-PEA) formation with LeAOS3 is presumably due to the occurrence of an oxyallyl diradical, a precursor of both cyclopropanone and cyclopentenone. In contrast, the yields of both the Favorskii-type product and *cis*-10-PEA are dramatically lower with ZmAOS. These findings allow us to propose that CYP74C AOSs such as LeAOS3 produce a tautomeric pair of allene oxide and oxyallyl, with a higher proportion of the latter. Equilibrium conversion of allene oxide and oxyallyl might also explain the appearance of the (10*Z*) isomer of allene oxide along with the ordinary (10*E*)-isomer. Back conversion of oxyallyl to allene oxide may be accompanied by a partial inversion of double-bond geometry.

These thoughts are supported by the following observations. Firstly, 10-oxo-PEA is quickly formed in parallel with allene oxide, 9,10-EOD and (9*R*)–α–ketol, as seen from the results of trapping experiments with LeAOS3 but not ZmAOS. In contrast, conversion of 13-HPOT via 12,13-EOT leads to α–ketol, trapping product, while the 12-oxo-PDA yield was relatively low. Secondly, keeping allene oxide 9,10-EOD (free acid) in hexane results in macrolactone formation via intramolecular nucleophilic substitution with the carboxylic group but no observable cyclization into 10-oxo-PEA. This indicates that the cyclization, if it ever occurred to a small extent, could not compete with the outrunning macrolactone formation.

### 3.4. Concluding Remarks

1. (9*R*)–α–ketol is a kinetically controlled LeAOS3 product in addition to the 9,10-EOD, as judged by the data of biphasic incubations followed by ethanol trapping. The α–ketol was detected at nearly equal yields with allene oxide ethanolysis product upon the very brief biphasic incubations. In contrast, ZmAOS (CYP74A19) produced predominantly the 9,10-EOD (detected as the ethanolysis product) upon the identical incubations, while α–ketol was only a minority.

2. The cyclopentenone *cis*-10-oxo-PEA was formed at nearly constant proportion to α–ketol (about 1:20), irrespective of incubation time and LeAOS3 concentration. No cyclopentenone was detected after the incubations with ZmAOS.

3. When the 9,10-EOD (free carboxylic acid) was prepared with LeAOS3 or ZmAOS and kept in hexane solution, it becomes macrolactones but does not exhibit any observable cyclization. Two macrolactones were detected, (12*Z*)-10-oxo-12-octadecene-11-olide and (12*Z*)-10-oxo-12-octadecene-9-olide.

4. The Favorskii-type product (Et/Me ester), formed via the ethanolysis of cyclopropanone, was detected after the short biphasic incubations of LeAOS3 with 9-HPOD followed by EtOH trapping. Its appearance indicates the presence of cyclopropanone, a short-lived valence tautomer of allene oxide (9,10-EOD), along with allene oxide itself. The yield of the Favorskii-type product after 9-HPOD incubations with ZmAOS was 50–100-times lower. The Favorskii-type product was also detected after both LeAOS3 and ZmAOS incubations with 13-HPOT, but not 13-HPOD. In addition to the Favorskii-type product, the cyclopentenone, *cis*-12-oxo-PDA, was detectable at a low yield after the brief incubations of both enzymes with 13-HPOT.

5. Allene oxides (Me) synthesized by LeAOS3 and ZmAOS were isolated and purified by NP-HPLC. ZmAOS specifically yielded the 10(*E*)-9,10-EOD, whereas CYP74C3 produced a roughly 4:1 mixture of 10(*E*) and 10(*Z*) isomers. The structure of both allene oxides was explored by ^1^H-NMR and ^1^H-^1^H-COSY spectroscopy, as well as the ^1^H-^1^H-NOESY, and ^1^H-^1^H-TOCSY, ^1^H-^13^C-HSQC and ^1^H-^13^C-HMBC for 10(*E*)-9,10-EOD synthesized by ZmAOS.

6. Oxyallyl diradical, a common intermediate of cyclopropanone and cyclopentenone formation, is supposed to be produced by LeAOS3 as a tautomeric form of allene oxide (9,10-EOD). This might explain the uncommon capability of LeAOS3 to produce appreciable yields of *cis*-10-oxo-PEA and the Favorskii-type product in contrast with ZmAOS. The appearance of a distinctive 10(*Z*) isomer of allene oxide with LeAOS3 may also be caused by the double-bond geometry inversion via the oxyallyl–allene oxide tautomeric equilibrium.

## 4. Materials and Methods

### 4.1. Materials

Linoleic acid and α-linolenic acids, as well as the soybean lipoxygenase type V, were purchased from Sigma. Silylating reagents and NaBH4 were purchased from Fluka (Buchs, Switzerland). (9*S*,10*E*,12*Z*)-9-Hydroperoxy-10,12-octadecadienoic acid (9-HPOD) was prepared by incubation of linoleic acids with the recombinant maize 9-lipoxygenase GeneBank: AAG61118.1) in Na-phosphate buffer (100 mM, pH 7.0) at 23 °C, under continuous oxygen bubbling. (9*Z*,11*E*,13*S*)-13-Hydroperoxy-9,11-octadecadienoic acid (13-HPOD) and (9*Z*,11*E*,13*S*,15*Z*)-13-hydroperoxy-9,11,15-octadecatrienoic acid (13-HPOT) were obtained via incubation of linoleic and α-linolenic acids, respectively, with the soybean lipoxygenase type V in Tris-HCl buffer (50 mM, pH 9.0) at 23 °C, under continuous oxygen bubbling. All hydroperoxides were purified by normal-phase HPLC.

### 4.2. Expression and Purification of Recombinant Enzyme

The open reading frame (ORF) of the maize AOS (ZmAOS, *CYP74A19*) was earlier cloned into the pET32 Ek/LIC vector (Novagen, Madison, WI, USA) to yield the target recombinant proteins with His-tags at N and C termini [21]. Recombinant plasmid pET-23a containing the ORF of the tomato AOS (LeAOS3, *CYP74C3*) was generously gifted by Prof. G. Howe (Michigan University, USA). The resulting constructions were transformed into the *Escherichia coli* host strain BL21 (DE3)pLysS (Novagen, Madison, WI, USA). The recombinant gene was expressed in host cells, as described before [32]. Purification of the His-tagged recombinant protein was performed using a Bio-Scale Mini Profinity IMAC (immobilized metal affinity chromatography) cartridge in the Bio-Rad NGC Discover 10 Chromatography System (Bio-Rad Laboratories, Moscow, Russia). The recombinant enzyme was eluted from the cartridges using 50 mM histidine. The homogeneity in the purified protein was confirmed by SDS-PAGE. The haemoprotein concentration was estimated using the pyridine haemochromogen assay [33].

### 4.3. Ethanol Trapping Experiments with 9-HPOD, 13-HPOT or 13-HPOD

The recombinant LeAOS3 or ZmAOS (25 µg, 50 µg, 100 µg, 150 µg or 200 µg) dissolved in 100 mM phosphate buffer (100 µL), pH 7.0, was extensively vortexed with 9-HPOD, 13-HPOT or 13-HPOD (100 µg) in hexane (4 mL) at 0 °C for 20–60 s. The water was quickly frozen at −79 °C, and the hexane phase was decanted and treated with ethereal diazomethane at 0 °C for 3 min. The predominant part of hexane was evaporated in vacuo and an excess of ice-cold ethanol was directly added to the solution. After 30 min at 23 °C, solvent was evaporated, and the dry residue was treated with a trimethylsilylating mixture pyridine – hexamethyldisilazane – trimethylchlorosilane (2:1:2, by volume). The products were subjected to GC-MS analyses.

### 4.4. Allene Oxide Isolation

The allene oxide 9,10-EOD (Me) was prepared as in trapping during incubations of ZmAOS and LeAOS3 with 9-HPOD followed by methylation with diazomethane and purification by NP-HPLC on LiChrosphere CN (5 μm) 30 × 4 mm column maintained at –15 °C, eluted with hexane/diethyl ether (97:3, by volume), flow rate 0.4 mL/min, using ultraviolet detection with Shimadzu SPD-M20 A diode array detector.

### 4.5. Separation and Purification of Products

The α-ketol (Me) preparations were purified firstly by RP-HPLC on Macherey–Nagel Nucleosil 5 ODS column (250 × 4 mm, 5 μm) using the solvent mixture methanol–water (linear gradient from 76:24 to 96:4, by volume) at a flow rate of 0.4 mL/min. Final purification was carried out by NP-HPLC on Macherey–Nagel Nucleodur 100–3 column (250 × 4.6 mm, 3 μm) eluted with hexane/isopropanol (99:1, by volume), flow rate 0.4 mL/min. Enantiomers of purified α-ketol methyl ester were separated on Chiralcel OB-H column (250 × 0.46 mm, 5 μm) with hexane/isopropanol 94:6 (by volume), flow rate 0.4 mL/min.

### 4.6. The Allene Oxide Conversions in an Aprotic Solvent

LeAOS3 and ZmAOS preparations (100 µg) dissolved in 100 mM phosphate buffer (100 µL), pH 7.0, were extensively vortexed with 9-HPOD (200 µg) in hexane (4 mL) at 0 °C for 60 s. The water was quickly frozen at −79 °C, and the hexane phase was decanted. The resulting allene oxide (free acid) preparation was allowed to stay in hexane solution for 20 h. Then, the products were treated with diazomethane, trimethylsilylated and subjected to GC-MS analyses.

### 4.7. Methods of Spectral Analyses

Products were analysed as Me/TMS derivatives by GC–MS. The GC–MS analyses were performed using a Shimadzu QP5050A mass spectrometer connected to a Shimadzu GC-17A gas chromatograph equipped with a Supelco MDN-5S (5% phenyl, 95% methylpolysiloxane)-fused capillary column (length, 30 m; ID 0.25 mm; film thickness, 0.25 μm). Helium at a flow rate of 30 cm/s was used as the carrier gas. Injections were made in split mode using an initial column temperature of 120 °C and injector temperature 230 °C. The column temperature was raised at 10 °C/min until 240 °C. The electron impact ionization (70 eV) was used. Most GC–MS analyses were carried out in full-scan mode. The NMR ^1^H, ^1^H-^1^H-COSY, ^1^H-^13^C-HSQC, ^1^H-^13^C-HMBC, ^1^H-^1^H-NOESY and ^1^H-^1^H-TOCSY spectra ([^2^H_14_]n-hexane) were recorded on a Bruker Avance III 600 spectrometer at 253 K.

## Figures and Tables

**Figure 1 ijms-24-02230-f001:**
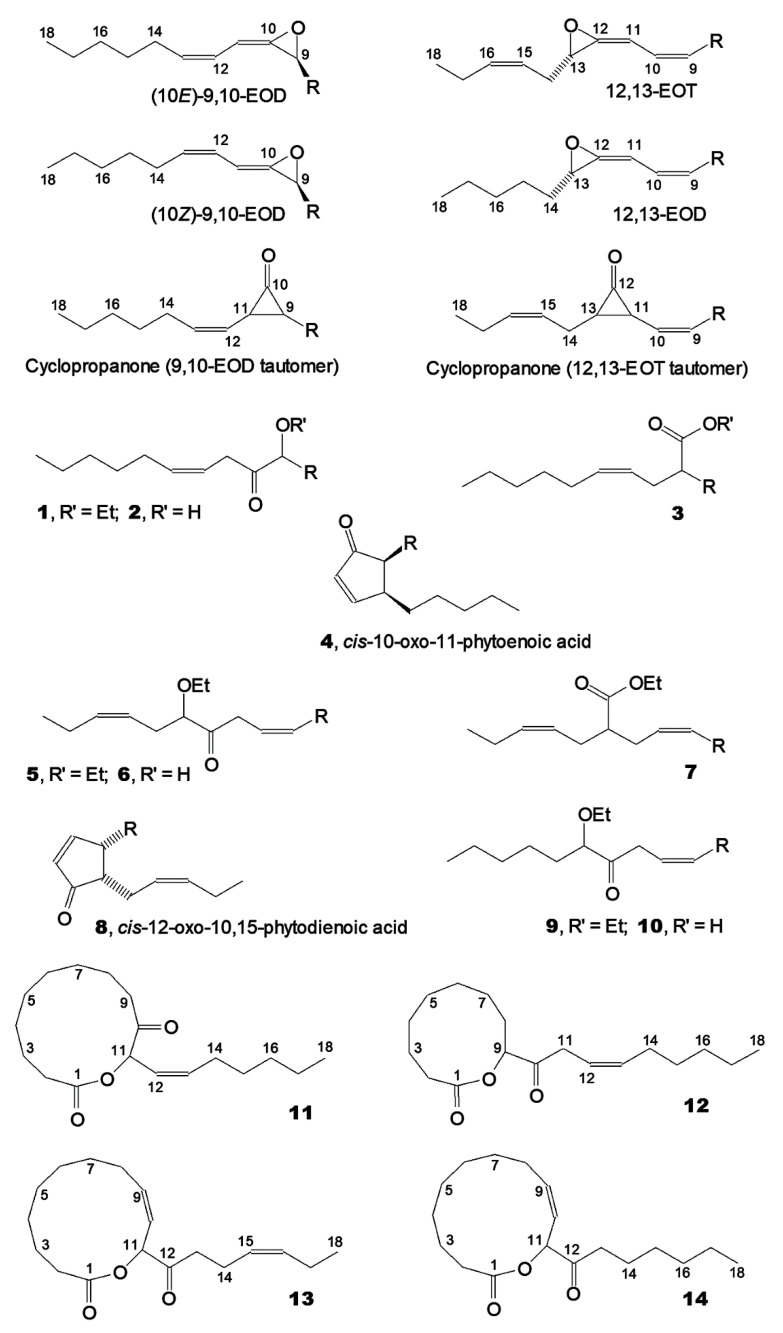
The structural formulae of the short-lived AOS products, allene oxides, their valence tautomers cyclopropanones and the products of their conversions detected in the present work. R = -(CH_2_)_7_COOH.

**Figure 2 ijms-24-02230-f002:**
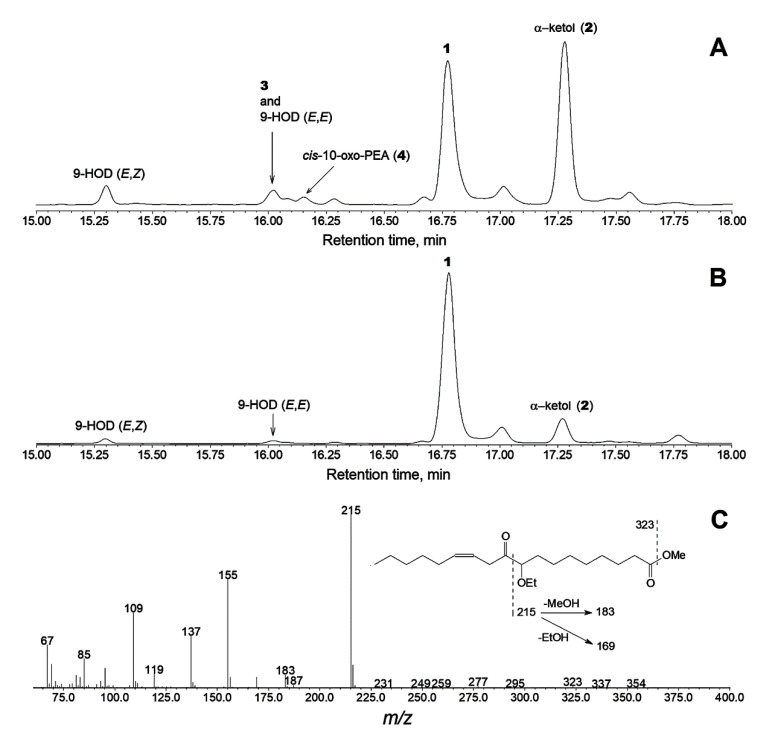
Products of ethanol trapping experiments after the brief incubations of 9-HPOD with LeAOS3 or ZmAOS. The total ion current GC–MS chromatograms of products (Me/TMS) of biphasic (hexane/water 40:1 by volume, 60 s, 0 °C) incubations of LeAOS3 (**A**) and ZmAOS (**B**) with 9-HPOD followed by (1) water freezing, (2) direct addition of ethanol excess to the hexane phase, (3) solvent evaporation, methylation and trimethylsilylation of products and (4) GC-MS analyses of product derivatives. (**C**), the electron impact mass spectrum of the trapping product **1**. Inset, the mass fragmentation scheme. The detailed experimental conditions are described in Materials and Methods.

**Figure 3 ijms-24-02230-f003:**
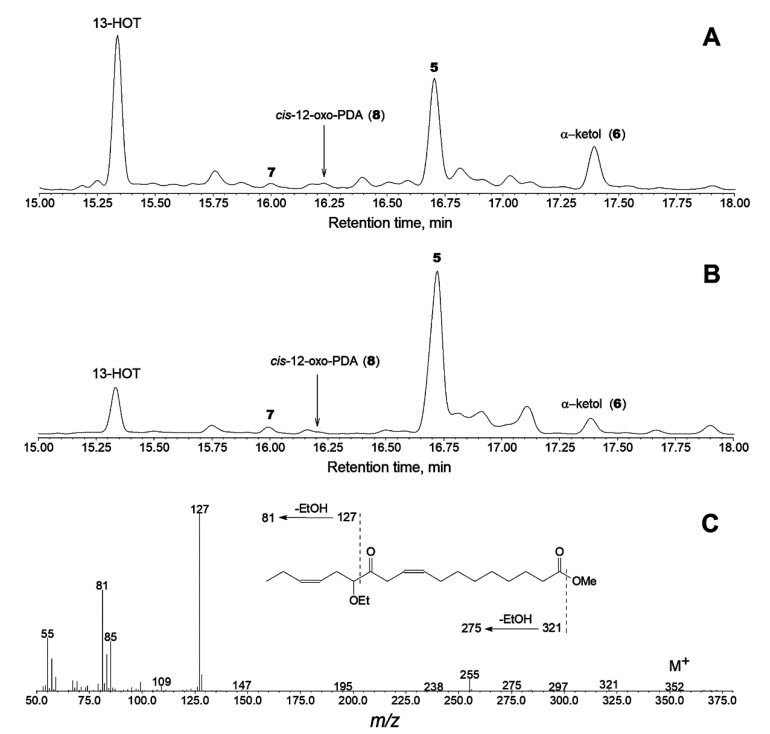
Products of ethanol trapping experiments after the brief incubations of 13-HPOT with LeAOS3 or ZmAOS. The total ion current GC–MS chromatograms of products (Me/TMS) of biphasic (hexane/water 40:1 by volume, 60 s, 0 °C) incubations of LeAOS3 (**A**) and ZmAOS (**B**) with 13-HPOT followed by (1) water freezing, (2) direct addition of ethanol excess to the hexane phase, (3) solvent evaporation, methylation and trimethylsilylation of products and (4) GC-MS analyses of product derivatives. (**C**) The electron impact mass spectrum of the trapping product **3**. Inset, the mass fragmentation scheme. The detailed experimental conditions are described in Materials and Methods.

**Figure 4 ijms-24-02230-f004:**
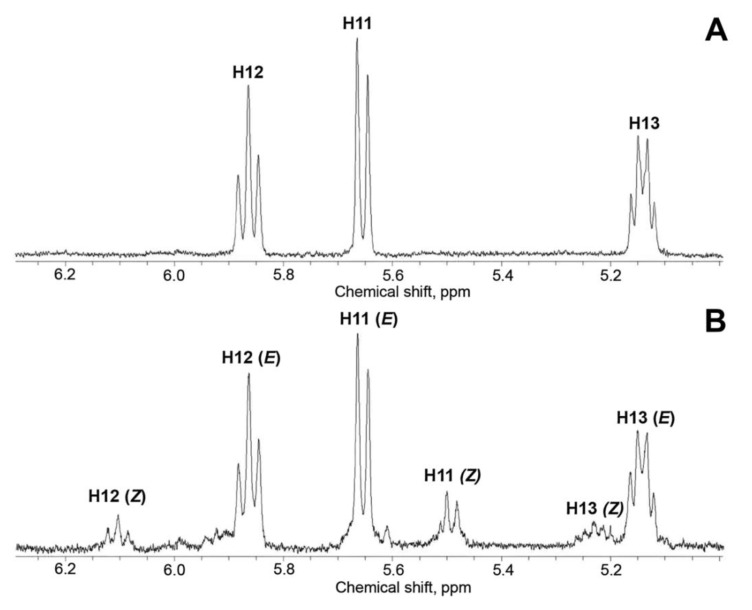
The partial ^1^H-NMR spectra (600 MHz, [^2^H_14_]n-hexane, 253 K) of allene oxide (9,10-EOD) Me ester preparations obtained with ZmAOS (**A**) and LeAOS3 (**B**).

**Figure 5 ijms-24-02230-f005:**
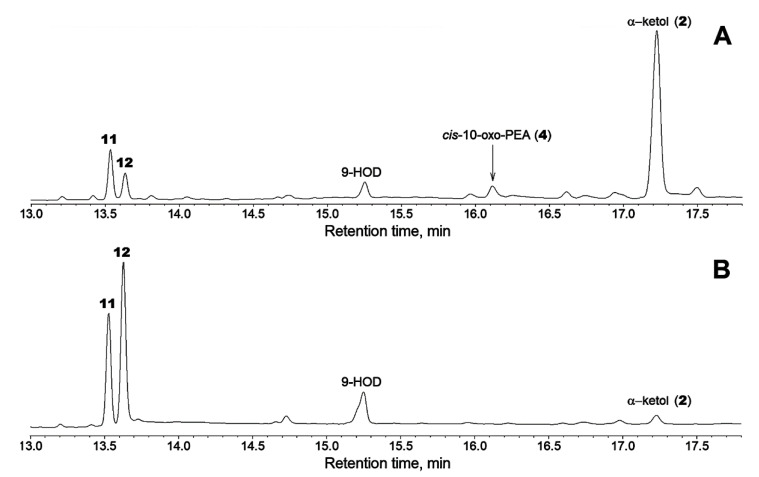
The GC-MS profiles of products (Me/TMS) formed when the hexane solution of isolated allene oxide 9,10-EOD (free acid) was kept for 2 h. The 9,10-EOD was preliminarily prepared by biphasic incubation of LeAOS3 (**A**) or ZmAOS (**B**) with 9-HPOD as described in Materials and Methods.

**Figure 6 ijms-24-02230-f006:**
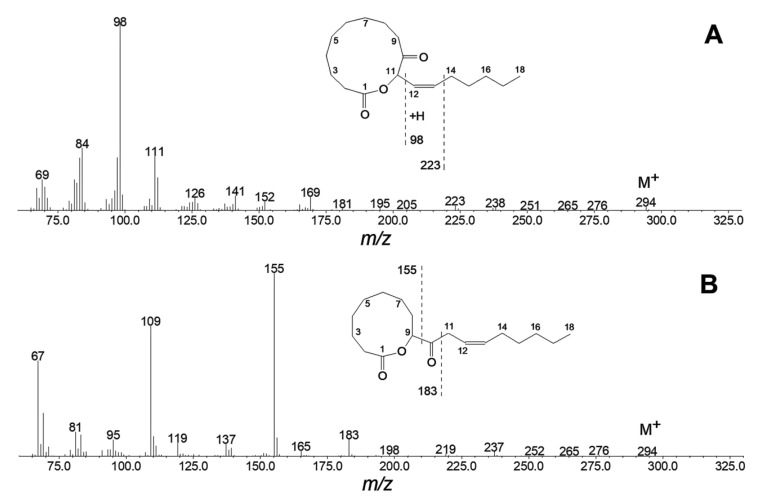
The electron impact mass spectra of products **11** (**A**) and **12** (**B**) formed when 9,10-EOD was kept for 2 h in hexane solution. Insets, the mass fragmentation schemes. The detailed experimental conditions are described in the Materials and Methods.

**Figure 7 ijms-24-02230-f007:**
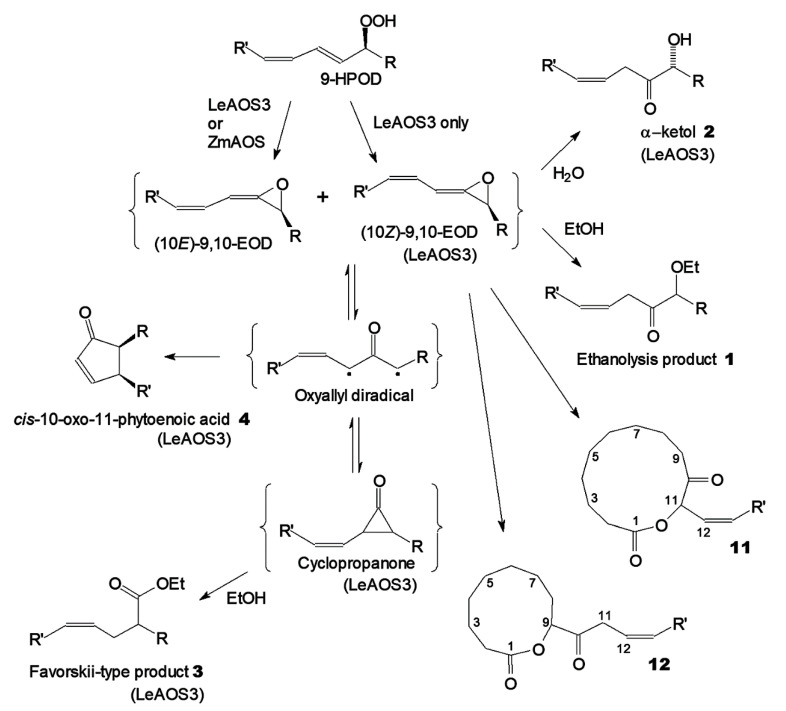
Scheme of 9-HPOD conversions to various products in the presence of LeAOS3 or ZmAOS followed by either ethanol trapping (all products except **11** and **12**) or keeping allene oxide in hexane (products **11** and **12**). R = -(CH_2_)_7_COOH; R′ = -(CH_2_)_4_Me. The inscription (LeAOS3) in brackets under the names of separate products means that these products were formed only or predominantly with LeAOS3, not with ZmAOS.

**Table 1 ijms-24-02230-t001:** The NMR spectral data for 9,10-EOD (Me ester) biosynthesized by ZmAOS (CYP74A19), [^2^H_14_]n-hexane, 253 K. The ^1^H-NMR (600 MHz), ^1^H-^1^H-COSY (600/600 MHz), ^1^H-^1^H-TOCSY (600/600 MHz), ^1^H-^13^C-HSQC (600/151 MHz), ^1^H-^13^C-HMBC (600/151MHz) and ^1^H-^1^H-NOESY (600/600 MHz) data are presented.

Position Number	^13^C Chemical Shifts (ppm); Functional Group	^1^H Chemical Shifts (ppm), Multiplicity, Coupling Constant (Hz)	2D NMR Correlations			
			COSY	TOCSY	HMBC	NOESY
1	172.73, COOMe					
2	34.21, CH_2_	2.19, t, 7.4 (H3)	H3	H3, H4	C1, C3, C4	H3
3	25.65, CH_2_	1.59, m	H2, H4	H2	C1, C2, C4	H2
4	30.17, CH_2_	1.31, m	H3		C3	
5	23.0–31.0, CH_2_	1.20–1.35, m				
6	23.0–31.0, CH_2_	1.20–1.35, m	H7	H9		
7	26.46, CH_2_	1.43, m	H6, H8	H9	C8	H9
8a	32.43, CH_2_	1.66, m	H7, H8b, H9	H7, H8b, H9, H11	C7, C9, C10	H8b, H9
8b		1.76, m	H7, H8a, H9	H7, H8a, H9	C7, C9, C10	H8a, H9
9	60.01, CH	3.46, dd, 5.5 (H8a), 4.5 (H8b)	H8a, H8b	H6, H7, H8a, H8b	C8, C10	H7, H8a, H8b, H11, H12
10	144.00, C					
11	86.39, CH	5.66, d 11.6 (H12)	H12	H8a, H12, H13, H14, H15	C9, C10, C12, C13	H9, H12, H14, H15
12	123.89, CH	5.87, dd (t-like), 11.6 (H11), 10.4 (H13)	H11, H13, H14	H11, H13, H14	C10, C11, C14	H9, H11, H13,
13	126.96, CH	5.14, dt, 10.4 (H12), 7.5 (H14)	H12, H14	H11, H12, H14, H15, H16, H18	C11, C14	H12, H14, H15
14	28.39, CH_2_	2.12, dt, 7.4 (H13), 7.4 (H15)	H12, H13, H15	H11, H12, H13, H15, H16, H18	C8?, C12, C13, C15	H11, H13, H15
15	30.64, CH_2_	1.39, m	H14	H11, H13		H14
16	30.49, CH_2_	1.20–1.35, m		H13		
17	23.74, CH_2_	1.32, m	H18			
18	14.67, CH_3_	0.89, t, 7.0 (H17)	H17	H11, H12, H13, H14	C16, C17	
(1)	50.98, COOMe	3.52, s			COOMe, C2, C3	

**Table 2 ijms-24-02230-t002:** The partial NMR spectral data for 10(*Z*)-isomer of allene oxide 9,10-EOD (Me ester) biosynthesized by LeAOS3 (CYP74A3), [^2^H_14_]n-hexane, 253 K. The ^1^H-NMR (600 MHz) and ^1^H-^1^H-COSY (600/600 MHz) data are presented.

Position Number	^1^H Chemical Shifts (ppm), Multiplicity, Coupling Constant (Hz)	2D NMR Correlations	
		COSY	TOCSY
11	5.49, d 11.3 (H12)	H12	H12, H13
12	6.11, dd (t-like), 11.3 (H11), 10.9 (H13)	H11	H11, H14
13	5.23, dt, 10.9 (H12), 7.5 (H14)		H11, H14

**Table 3 ijms-24-02230-t003:** The relative abundance of different products after the brief biphasic incubations of LeAOS3 and ZmAOS with fatty acid hydroperoxides followed by ethanol trapping. Fulfilled boxes, major products; partly filled boxes, minor products; empty boxes, undetected products.

Enzyme	Substrate	Trapping Product	α–Ketol	Favorskii-Type Product	Cyclopentenone
LeAOS3	9HPOD	qqqqqqййййй	qqqqqqййййй	Qq	qq
ZmAOS	9HPOD	qqqqqqййййй	qq		
LeAOS3	13HPOT	qqqqqqййййй	qqqqqqййййй	Qq	
ZmAOS	13HPOT	qqqqqqййййй	qq	Qq	
LeAOS3	13HPOD	qqqqqqййййй	qqqqqqййййй		
ZmAOS	13HPOD	qqqqqqййййй	qq		

## Data Availability

Not applicable.

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
