# Peer review of "Distinct Mechanistic Behaviour of Tomato CYP74C3 and Maize CYP74A19 Allene Oxide Synthases: Insights from Trapping Experiments and Allene Oxide Isolation"

_ijms, 2023, doi:10.3390/ijms24032230_

Round 1
Reviewer 1 Report
Authors used novel biphasic approach to trap the nonpolar allene oxide intermediate produced by CYP74A(ZmAOS) or CYP74C(LeAOS3) and identified structures of trapped products by GCMS and NMR. Spectroscopic data are consistent with the corresponding structural identifications. Results are quite interesting.
Comments:
1. Table 3 shows the relative abundance of biphasic incubations of ZmAOS and LeAOS3 with 9-HPOD, 13-HOPD, 13-HPOT as substrates. Authors claim that the anchimeric assistance from b,g double bond play critical role for the relative abundance from the result obtained by 13-HPOT. However, data with 9-HPOT as a substrate is missing. In fact, 9-HPOT is easily available and can be prepared by 9-LOX(CaLOX).
2. Differences between ZmAOS and LeAOS3 in Table 3 need to be explained in terms of mechanistic viewpoints of enzymatic mechanisms associated with ZmAOS and LeAOS3. Are there any correlations between the active site structure and the product distributions? More detailed information on ZmAOS and LeAOS3 are needed in Introduction section.
3. In Fig 4, ZmAOS produce a single geometrical isomer, however, mixture of isomers (4:1) are produce from LeAOS3. What is the reason for this difference?
4. If the product distribution data in the biphasic incubation experiments can compared with kinetic parameters of ZmAOS and LeAOS3.
5. When the 9-path in Fig. 2 is compared with the 13-path in Fig 3 are compared, larger amount of 13-HOT than 9-HOD is produced. This needs to be explained and Table 3 should include the relative abundance of 13-HOT and 9-HOD.
Author Response
Answers to the Reviewers comments of Reviewer 1
POINT 1. Table 3 shows the relative abundance of biphasic incubations of ZmAOS and LeAOS3 with 9-HPOD, 13-HOPD, 13-HPOT as substrates. Authors claim that the anchimeric assistance from b,g-double bond play critical role for the relative abundance from the result obtained by 13-HPOT. However, data with 9-HPOT as a substrate is missing. In fact, 9-HPOT is easily available and can be prepared by 9-LOX(CaLOX).
ANSWER: Indeed, no experiments with 9-HPOT were done. However, one should note that 9-HPOT and the related allene oxide 9,10-EOT are irrelevant in connection to anchimeric assistance. Both 9-HPOT and 9,10-EOT lack the beta,gamma-double bond at the oxirane required for this effect. Thus, only the 9-hydroperoxide of gamma-linolenic acid (gamma-9-HPOT) would be relevant. Previously we performed the experiments on gamma-9-HPOT conversion by flax and maize AOSs, as well as the LeAOS3 [Grechkin et al., 2011]. The obvious effects of the assistance of both cyclopentenone and cyclopropanone formation (as judged by the Favorskii-type product) were observed. Neither cyclopentenone nor the Favorskii-type product was formed from 9-HPOD in the presence of flax or maize AOSs [Grechkin et al., 2011].
One should note in addition that the effect of anchimeric assistance is mentioned in relation to the literature data. The conditions of the brief incubations used in the present work are not preferable to observing the formation of cyclopentenones (even with 13-HPOT), except in the case of LeAOS3 incubations with 9-HPOD.
The formation of the Favorskii-type product from 13-HPOT and loss of its formation from 13-HPOD were described [Grechkin et al., 2011].
POINT 2. Differences between ZmAOS and LeAOS3 in Table 3 need to be explained in terms of mechanistic viewpoints of enzymatic mechanisms associated with ZmAOS and LeAOS3. Are there any correlations between the active site structure and the product distributions? More detailed information on ZmAOS and LeAOS3 are needed in Introduction section.
ANSWER: The data on the difference in alpha-ketol formation (Table 3) are more confirmatory than original. Capability of potato CYP74C10 (analogous to CYP74C3) to produce 9(R)-alpha-ketol from 9-HPOD at a high extent of stereospecificity was originally observed by Mats Hamberg (2000). Our data suggest that 9(R)-alpha-ketol is a kinetically controlled product synthesized by LeAOS3 along with allene oxide. However, the discussion of the enzymological background of these phenomena seems premature due to the loss of X-ray data for any CYP74C enzyme. The 3D structure is still available only for AtAOS (CYP74A1).
Some additional clarifications are added to the Introduction.
POINT 3. In Fig 4, ZmAOS produce a single geometrical isomer, however, mixture of isomers (4:1) are produce from LeAOS3. What is the reason for this difference?
ANSWER: The allene oxide produced by LeAOS3 was first identified as a mixture of (Z) and (E) isomers by Brash et al. (2013). This difference between CYP74C and CYP74A AOSs was attributed to enzyme specificity. We have the same opinion.
POINT 4. If the product distribution data in the biphasic incubation experiments can compared with kinetic parameters of ZmAOS and LeAOS3.
ANSWER: The overall activity of LeAOS3 and ZmAOS towards 9-HPOD as well as the kinetics were comparable. The product distribution was dependent on the distinct specificities of two enzymes. The obtained data suggested a shift of tautomeric equilibrium between allene oxide and oxyallyl in favour of the latter in the case of LeAOS3. This assumption is consistent with the elevated yields of cyclopentenone 10-oxo-PEA, the Favorskii-type product, as well as the appearance of 10(Z)-allene oxide. All these peculiarities are distinctive for LeAOS3.
POINT 5. When the 9-path in Fig. 2 is compared with the 13-path in Fig 3 are compared, larger amount of 13-HOT than 9-HOD is produced. This needs to be explained and Table 3 should include the relative abundance of 13-HOT and 9-HOD.
ANSWER: The abundance of hydroxy acids in GC-MS profiles reflects the amount of remaining unconverted hydroperoxide. So, a large peak of 13-HOT was present because 13-HPOT is a poorer LeAOS3 substrate than 9-HPOD.

Reviewer 2 Report
In this manuscript, Grechkin and coworker described the mechanistic behavior of two allene oxide synthases, tomato CYP74C3 and maize CYP74A19. Considering 9-HOPD, 13-HPOT and 13-HPOD as substrates, they performed multiple trapping experiments. Incubation of these substrates with AOSs delivered different products, which were carefully isolated and characterized with GC-MS and NMR. Considering the novelty and usefulness of these results, this referee recommends the manuscript for publication after addressing following comments.
1. Incubation of 9-HOPD, 13-HPOT and 13-HPOD with AOSs provided many closely related products. All products should be carefully labelled with numbers and referenced to the main text. This will help readers to follow.
2. A detailed scheme with structures for 13-HPOT and 13-HPOD conversions to various products should be added to the manuscript, as performed for substrate 9-HPOD (Fig 7).
3. Stereochemistry for α-ketol (top right compound in the Fig 7) should be double checked. Does ethanolysis product of 9,10-EOD provide the mix of enantiomers? Otherwise, stereochemistry should be mentioned in the structure (Fig 7, middle right structure).
4. 1H, 13C NMR diagram and 2D NMR correlation diagram for allene oxide 9,10-EOD should be added to the supporting information.
5. This work is a detailed investigation of mechanistic behavior of allene oxide synthase, and a significant amount of work has been performed to characterize the products. In addition to that, authors should provide insights on the importance and application of this work. A short discussion in the Introduction should be enough.
Author Response
Answers to the comments of Reviewer 2
In this manuscript, Grechkin and coworker described the mechanistic behavior of two allene oxide synthases, tomato CYP74C3 and maize CYP74A19. Considering 9-HOPD, 13-HPOT and 13-HPOD as substrates, they performed multiple trapping experiments. Incubation of these substrates with AOSs delivered different products, which were carefully isolated and characterized with GC-MS and NMR. Considering the novelty and usefulness of these results, this referee recommends the manuscript for publication after addressing following comments.
POINT 1. Incubation of 9-HOPD, 13-HPOT and 13-HPOD with AOSs provided many closely related products. All products should be carefully labelled with numbers and referenced to the main text. This will help readers to follow.
ANSWER: Conventionally, compounds are numbered in the order of their mention in the text. The numbering of the short-lived intermediates (allene oxides and cyclopropanones) would lead to a confusing order of numbering. In accordance with the recommendation, we numbered the alpha-ketols and 12-oxo-PDA in addition. All other products are renumbered accordingly.
POINT 2. A detailed scheme with structures for 13-HPOT and 13-HPOD conversions to various products should be added to the manuscript, as performed for substrate 9-HPOD (Fig 7).
ANSWER: The schemes of 13-HPOT and 13-HPOD are drawn and added to the Supplementary Information.
POINT 3. Stereochemistry for α-ketol (top right compound in the Fig 7) should be double checked. Does ethanolysis product of 9,10-EOD provide the mix of enantiomers? Otherwise, stereochemistry should be mentioned in the structure (Fig 7, middle right structure).
ANSWER: Indeed, the alpha-ketol stereochemistry in Fig. 7 was wrong. This is corrected.
We did not study the stereochemistry of the ethanolysis product. Presumably it should be racemic.
POINT 4. 1H, 13C NMR diagram and 2D NMR correlation diagram for allene oxide 9,10-EOD should be added to the supporting information.
ANSWER: The 1H,13C-HSQC, 1H,13C-HMBC, and 1H,1H-COSY spectra are drawn and added to the Supplementary Information.
POINT 5. This work is a detailed investigation of mechanistic behavior of allene oxide synthase, and a significant amount of work has been performed to characterize the products. In addition to that, authors should provide insights on the importance and application of this work. A short discussion in the Introduction should be enough.
ANSWER: A sentence summarizing the novelty and significance of the present work is added at the end of the Introduction.
